# Global water gaps under future warming levels

Lorenzo Rosa [1] & Matteo Sangiorgio [2]

Understanding the impacts of climate change on water resources is crucial for developing effective adaptation strategies. We quantify "water gaps", or unsustainable water use – the shortfall where water demand exceeds supply, resulting in scarcity. We quantify baseline and future water gaps using a multi-model analysis that incorporates two plausible future warming scenarios. The baseline global water gap stands at 457.9 km³/yr, with projections indicating an increase of 26.5 km³/yr (+5.8%) and 67.4 km³/yr (+14.7%) under 1.5 °C and 3 °C warming scenarios, respectively. These projections highlight the uneven impact of warming levels on water gaps, emphasizing the need for continued climate change mitigation to alleviate stress on water resources. Our results also underscore the unequal adaptation needs across countries and basins, influenced by varying warming scenarios, with important regional differences and model variability complicating future projections. Robust water management strategies are needed to tackle the escalating water scarcity caused by global warming.

Water scarcity is one of the defining challenges of the 21st century. Water scarcity occurs when the consumption of freshwater exceeds its availability, creating an imbalance[1,2]. This condition signifies a complex state of human deprivation, where access to affordable and safe water to meet societal needs is inadequate, or where these needs are met at the expense of the environment[3,4]. While water scarcity can affect entire regions, the most severe consequences are borne by the most vulnerable and impoverished populations, underscoring the important influence of economic and institutional factors in determining water scarcity[5–7].

In many regions of the world, water demand surpasses the sustainable supply from rivers and aquifers[8–11]. Currently, half of the world's irrigated agriculture[12] and about 4 billion people reside in regions experiencing water scarcity for at least one month per year[13,14]. Climate change is disrupting precipitation and evapotranspiration patterns[15,16], further aggravating water scarcity[14,17–20]. The increasing demand for water, driven by population growth and urbanization[21,22], pollution[23], expanded irrigated agriculture[24–26], and greater industrial use[27], exacerbates water scarcity[28]. Under global warming, this fragile balance between supply and demand is likely to worsen, leading to a future where water resources struggle to meet growing societal and environmental needs[14,29–31]. Consequently, many areas face a widening water gap, which threatens not only economic development and societies[32] but also the health of aquatic ecosystems[33].

First introduced in 2009[34] and widely utilized since, water gaps indicate the difference between renewable water availability and water consumption within a specific region. A water gap occurs when, in any given month, water consumption surpasses local available renewable water resources. These gaps highlight unsustainable water use, leading to the depletion of groundwater, rivers, lakes, and environmental flow reserves. Such situations signal critical shortages where water resources cannot meet the demands of populations, agriculture, industry, energy production, and ecosystems, potentially resulting in severe social, economic, and environmental consequences. Quantifying water gaps provides decision-makers with vital information for effective water scarcity management.

Past research has primarily concentrated on quantifying groundwater depletion[8,11,35–37] and environmental flow reduction[3,12,30,38–40] on a global scale. Other studies have explored groundwater depletion and unsustainable water use at regional levels[41–46]. However, comprehensive global assessments using multi-model and multi-scenario analyses to evaluate both historical and future water gaps remain lacking. These

¹Biosphere Sciences and Engineering, Carnegie Institution for Science, Stanford, CA, USA. ²Department of Electronics, Information and Bioengineering, Politecnico di Milano, Milan, Italy. ✉e-mail: lrosa@carnegiescience.edu

are missing elements that are essential for informed decision-making and effective adaptation to water scarcity amid uncertainty. This research gap underscores the need for analyses that can guide policy and bolster resilience in the context of climate variability and growing water demand.

Here, we quantify baseline and future water gaps using a multi-model analysis that considers two plausible future warming scenarios to assess the uncertainty of projections. We quantify water gaps worldwide at a 30-arc minute resolution (50 km at the Equator) under baseline climate conditions and 1.5 °C and 3 °C warming scenarios, relative to the preindustrial era[24]. The years 2001–2010 serve as the baseline period for our analysis, enabling us to assess changes relative to recent historical conditions observed. A 3 °C warming represents a plausible level of global warming expected by the end of the century under current policies, whereas a 1.5 °C warming represents the target set in the Paris Agreement[24]. To capture a range of potential outcomes, we utilize outputs from five distinct climate models from the Coupled-Model Intercomparison Project (CMIP6) archive[47] to evaluate water gaps under both baseline and future warming conditions. First, using a hydrological model[1,4,12,24,48], we quantify renewable water availability from runoff estimates from CMIP6 climate outputs. Second, by comparing water availability and consumption for each warming scenario and climate model, we quantify water gaps at the pixel-level worldwide. Third, we present results at the major hydrological basin- and country-scale considering variations in water gaps among each of the five climate models considered in this study. We acknowledge the inherent variability in climate models and emphasize the importance of assessing and incorporating this uncertainty into our projections. By aggregating results at the hydrological basin- and country-scale, we capture both local and regional variations in water gaps, which are essential for designing targeted adaptation measures.

Our study uses publicly available, widely adopted, and validated climate and hydrological model outputs to quantify water gaps under baseline and future warming scenarios. The novelty of our approach lies in its multi-model and multi-scenario analysis, which accounts for groundwater depletion and the depletion of surface water and environmental flows, thus estimating water needs for ecosystems. By assessing water deficits during the baseline period (2001–2010) and under two future warming levels, our work provides valuable insights for water resource planning and adaptation strategies. It identifies varying adaptation needs, highlights key trends and regional vulnerabilities, and pinpoints countries and river basins where water gaps are projected to change substantially, supporting targeted adaptation measures.

## Results
### Geographic distribution and climate change impact on water gaps

Figure 1 illustrates the geographic variation of water gaps worldwide, showing the average water gaps among the five climate models considered in this study (model-specific results are available in Supplementary Figs. 1–5).

Water gaps, though regional, occur on every continent, highlighting regions where the mismatch between water availability and consumption is most pronounced. Accounting for ~90% of societal water consumption, irrigated agriculture dominates humanity's water footprint[25]. Major irrigation districts with substantial water gaps include California's Central Valley, the US High Plains, Central Chile, the Iberian Peninsula, Saudi Arabia, the Tigris-Euphrates River system, the Aral Sea Basin, the Indo-Gangetic Plains of India and Pakistan, the North China Plain, and Australia's Murray-Darling Basin (Fig. 1).

Global warming is increasingly exacerbating water gaps. Figure 1b, c depicts changes in water gaps relative to the baseline under 1.5 °C and 3 °C warmer climate scenarios. Regions currently experiencing water gaps are expected to face more severe conditions under

1.5 °C warming, with even worse outcomes at 3 °C warming. This trend is particularly evident in the eastern US, Chile, the Mediterranean region, South and East India, and the North China Plain. Additionally, some regions that were relatively unaffected in the baseline climate, such as Italy, Madagascar, and some US states on the East Coast (North Carolina and Virginia) and in the Great Lakes region (Wisconsin, Minnesota, Illinois), are projected to see worsening conditions (Fig. 1). Interestingly, Saudi Arabia is projected to experience decreased water scarcity under the 1.5 °C warming scenario, but substantial increases in water gaps under the 3 °C warming scenario (Fig. 1). Regions in Western North America, Africa's Sahel, Northwest India, central China, and Central Asia are projected to experience reduced water gaps (Fig. 1).

### Country-specific water gaps and model variability

The analysis was repeated by aggregating the average pixel-level water gaps at the country and global scale (Fig. 2 and Table 1). Figure 3 displays the volumes of water gaps per country and their variability among the five climate models considered in this study.

Under baseline conditions, our results show that the global water gap is 457.9 km³/yr (Table 1). In a 1.5 °C warmer climate, global water gaps are expected to increase to 484.4 km³/yr (+26.5 km³/yr) with respect to the baseline climate, with uncertainty among the five climate models ranging between 454.0 km³/yr (-3.9 km³/yr) and 514.3 km3/yr (+56.4 km³/yr) (Table 1). In a 3 °C warmer climate, global water gaps are expected to increase to 525.3 km³/yr (+67.4 km³/yr) with respect to the baseline climate, with uncertainty among the five climate models ranging between 495.9 km³/yr (+38.0 km³/yr) and 546.1 km³/yr (+88.2 km³/yr) (Table 1).

Under the baseline climate, the largest water gaps are found in India (124.3 km³/yr), the United States (53.8 km³/yr), Pakistan (35.8 km³/yr), Iran (35.0 km³/yr), and China (27.2 km³/yr) (Figs. 2a and 3). India is projected to experience the most important increase in water gaps under warming scenarios. In a 1.5 °C warmer climate, India will have an additional 11.1 km³/yr water gap (Figs. 2a and 3). China is the second country with the largest increase in water gaps (4.1 km³/yr), followed by Pakistan (2.4 km³/yr), the United States (2.3 km³/yr), Spain (1.6 km³/yr), and Türkiye (1.1 km³/yr) (Figs. 2a and 3).

Different warming levels show uneven changes in water gaps. In a 3 °C warmer climate, India still presents the largest water gap increase (17.2 km³/yr) compared to baseline conditions (Figs. 2a and 3), followed by Pakistan (11.7 km³/yr), the United States (7.4 km³/yr), China (7.3 km³/yr), Spain (3.5 km³/yr), and Türkiye (2.2 km³/yr) (Fig. 3). An increase in water gaps greater than 1 km³/yr is also projected for Iran (2.0 km³/yr), Saudi Arabia (1.5 km³/yr), Italy (1.3 km³/yr), Bangladesh (1.3 km³/yr) and Morocco (1.3 km³/yr) (Fig. 3). Interval bars in Fig. 3 depict the variations in water gaps based on our multi-model analysis. This analysis highlights large interval bars in the logarithmic scale (see, for instance, Pakistan and Bangladesh in a 3 °C climate; China in a 1.5 °C climate; Indonesia and Nigeria in both cases), representing a high variability in water gap projections (compared to the baseline water gap) due to the variability among the climate models (Fig. 3).

Global warming will alter precipitation patterns, leading to a reduction in water gaps in some countries. In Nigeria, water gaps are projected to decrease from 1.2 km³/yr under the baseline climate to 0.8 km³/yr and 0.6 km³/yr under 1.5 °C and 3 °C warming, respectively (Figs. 2b, c and 3). Similarly, in the Philippines, water gaps will decrease from 2.4 km³/yr under the baseline climate to 2.0 km³/yr and 2.3 km³/yr under 1.5 °C and 3 °C warming, respectively (Figs. 2b, c and 3). Other countries expected to see decreases in water gaps under warming include Sudan, Vietnam, Uzbekistan, the United Arab Emirates, Tajikistan, and Ethiopia (Figs. 2b, c and 3). Although global warming will reduce water gaps in these countries, the gaps will persist but with lesser magnitude.

For each country, we also assessed the agreement among climate models regarding changes in water gaps. Figure 2d, e

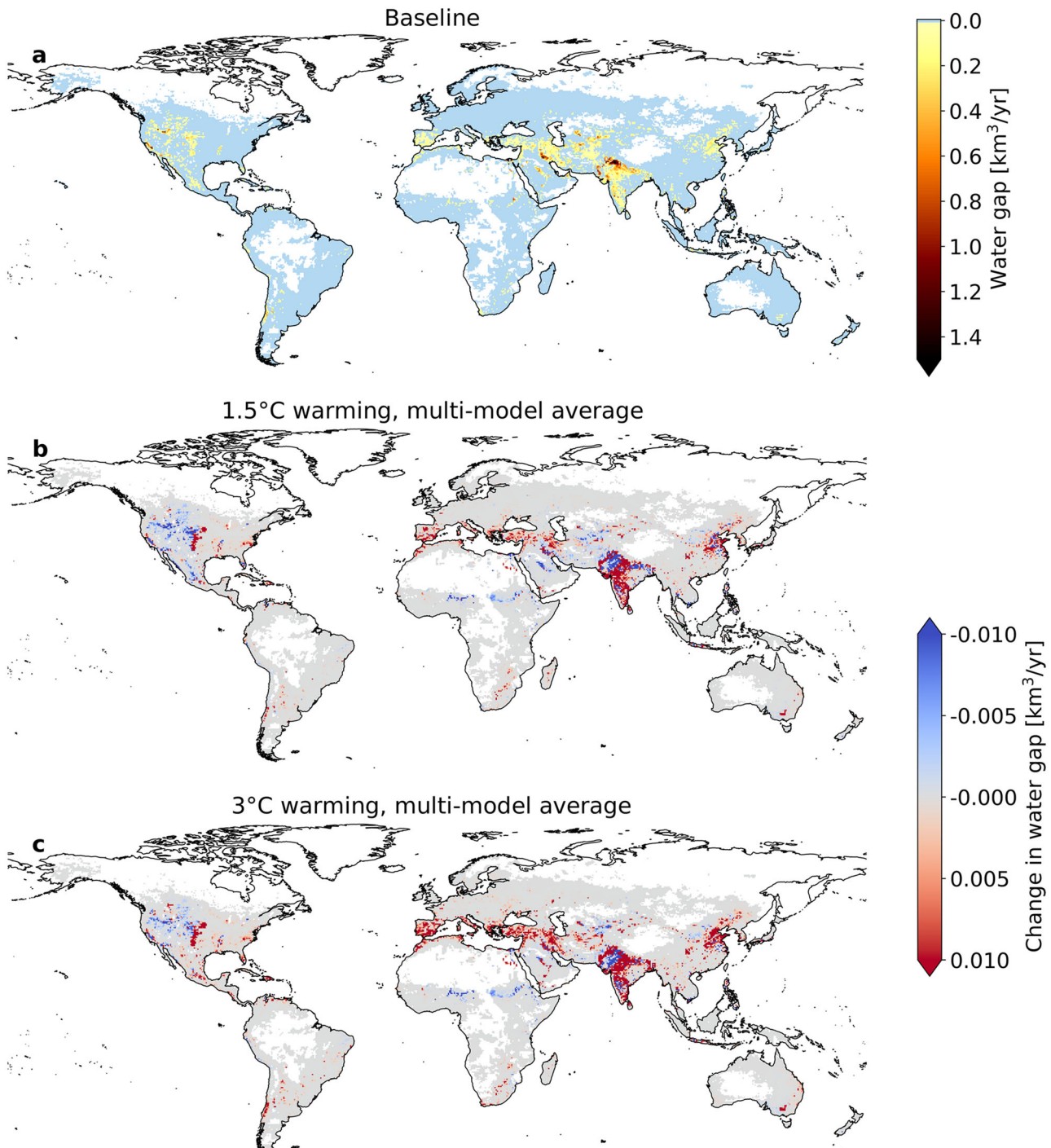

**Fig. 1 | Global water gaps under baseline and warming climates. a** Water gaps for the baseline period (2001–2010), and **b**, **c** changes in water gaps in a 1.5 °C and 3 °C warmer climate. The maps show the average among the five climate models considered in this study; model-specific water gaps are presented in Supplementary Figs. 1–5.

highlight countries where at least three of the five climate models concur in the direction of change in water gaps. Under warming scenarios, climate models generally agree on a widespread exacerbation of water gaps. However, some countries, including Canada, Mexico, Saudi Arabia, Nigeria, Sudan, Thailand, Vietnam, and the Philippines, are expected to see a decrease in water gaps under 1.5 °C warming (Fig. 2d). Under 3 °C warming, only Nigeria, Niger, Chad, Sudan, Ethiopia, Vietnam, and the Philippines will show a decrease in water gaps across models, though these reductions will be modest (Fig. 2e).

## Water gap trends by major hydrological basins

The analysis was repeated by aggregating the pixel-level water gaps at the scale of major hydrological basins worldwide (Fig. 4a–c). Figure 5 shows the volumes of water gaps per major hydrological basin and their variability among the five climate models considered in this study.

Under baseline climate conditions, the largest water gaps are found in the Ganges-Brahmaputra (56.1 km³/yr), Sabarmati (52.6 km³/yr), Tigris-Euphrates (34.1 km³/yr), Indus (28.7 km³/yr), and Nile River basins (22.2 km³/yr) (Figs. 4a and 5). Under 1.5 °C warming

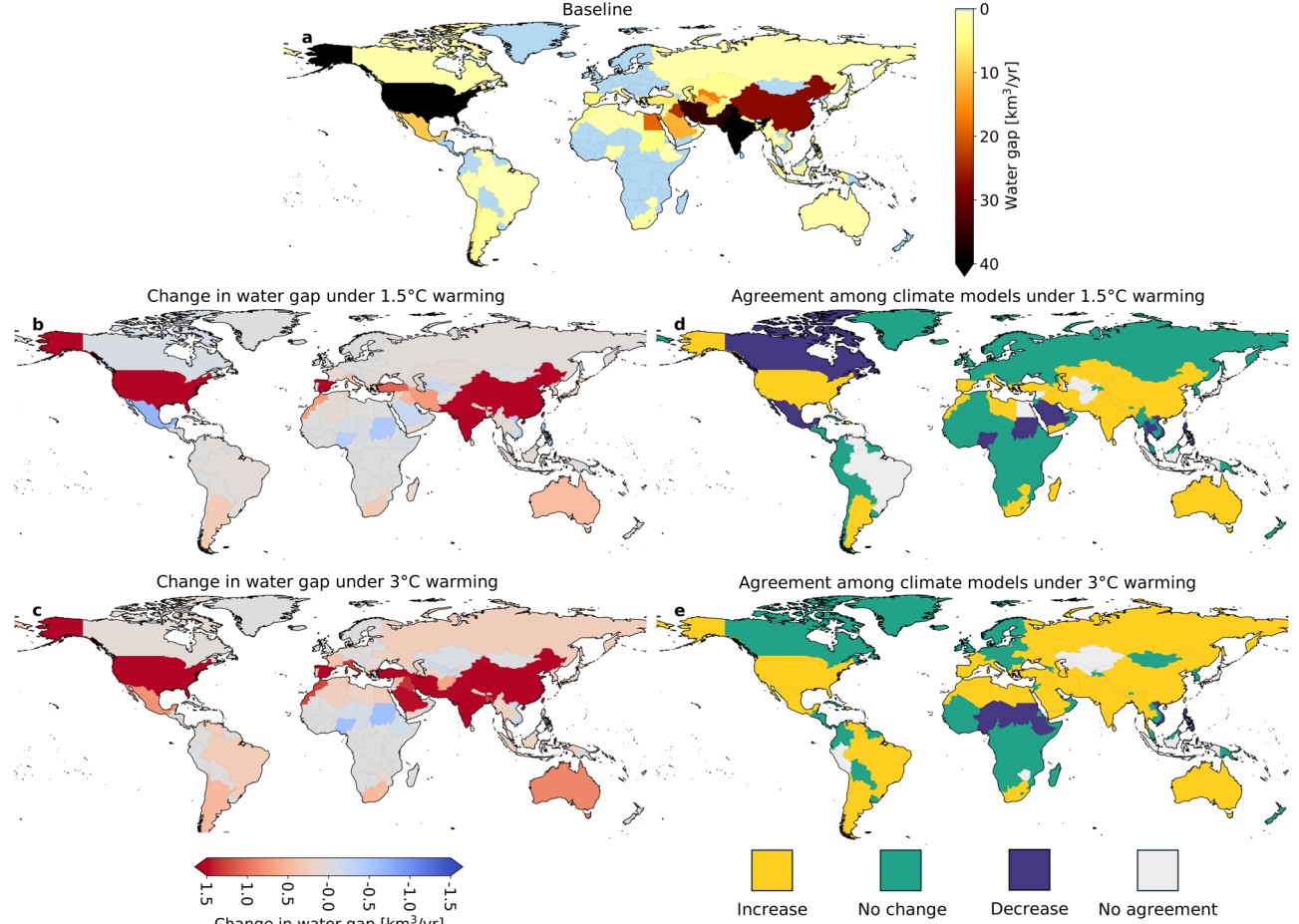

**Fig. 2 | Country-specific water gaps. a** Water gaps for the baseline period (2001–2010), and **b**, **c** variations in water gaps in a 1.5 °C and 3 °C warmer climate. The maps show the multi-model average among the five climate models considered in this study. Agreement among climate models regarding trends in water gap variations in a 1.5 °C (**d**) and 3 °C (**e**) warmer climate. Exacerbation of water gaps is shown in yellow, with no change in green and a decrease in water gaps in blue. Gray basins indicate a lack of agreement among climate models.

**Table 1 | Climate model-specific water gaps aggregated at the global scale under baseline and future warming levels**

| Baseline (km³/yr) | Warming Levels | | |
|---|---|---|---|
| | **Climate Model** | **1.5 °C (km³/yr)** | **3 °C (km³/yr)** |
| 457.9 | IPSL-CM6A-LR | 454.0 | 546.1 |
| | UKESM1-0-LL | 492.6 | 495.9 |
| | GFDL ESM4 | 490.0 | 539.1 |
| | MPI-ESM1.2-HR | 514.3 | 525.0 |
| | MRI-ESM2.0 | 471.2 | 520.4 |
| | Multi-model average | 484.4 | 525.3 |

Detailed model-specific and scenario-specific results per country and basin are presented in Supplementary Table 1, 2 and in Supplementary Data.

conditions, water gaps are expected to increase the most in the Ganges-Brahmaputra (5.6 km³/yr), Godavari (2.5 km³/yr), and Mississippi-Missouri (2.4 km³/yr) river basins, while water gaps are expected to decrease in the Sabarmati (1.4 km³/yr), Columbia and Northwestern United States (0.7 km³/yr), and Nile (0.6 km³/yr) basins (Figs. 4b, c and 5).

In a 3 °C warmer climate, the Ganges-Brahmaputra basin still presents the largest water gap increase (11.8 km³/yr) compared to baseline conditions, followed by the Indus (8.4 km³/yr), Mississippi-Missouri (5.5 km³/yr), China Coast (2.7 km³/yr), Godavari (2.6 km³/yr), and Tigris-Euphrates (2.6 km³/yr) basins (Fig. 5). Interval bars in Fig. 5

depict the variations in water gaps based on our multi-model analysis. This analysis highlights large interval bars in the logarithmic scale (representing a high variability in water gap projections compared to the baseline water gap) for Ziya He, China Coast, Gulf Coast, and Java-Timor basins (Fig. 5).

Under 1.5 °C warming, water gaps in the Sabarmati Basin are projected to decrease from 52.6 km³/yr under baseline conditions to 51.2 km³/yr (Figs. 4b, c and 5). Similarly, in the Nile basin, water gaps will decrease from 22.2 km³/yr under baseline conditions to 21.6 km³/yr and 21.5 km³/yr under 1.5 °C and 3 °C warming, respectively (Figs. 4b, c and 5).

For each major hydrological basin, we also evaluated the agreement among climate models regarding trends in water gap variations. Figure 4d, e highlight basins where at least three of the five climate models concur with the direction of change in water gaps. Climate models agree that water gaps are expected to increase in most hydrological basins but decrease in the Nile, Columbia, and North-western United States, North America's Colorado, Great Basin, Niger, and Lake Chad basins (Fig. 4d, e).

## Discussion

Our findings reveal that under baseline climate conditions, important water gaps already exist, amounting to 457.9 km³ per year, highlighting substantial challenges for water sustainability. Addressing these existing water gaps is crucial, as doing so could likely enhance adaptability to future changes in most river basins. Water gaps are projected

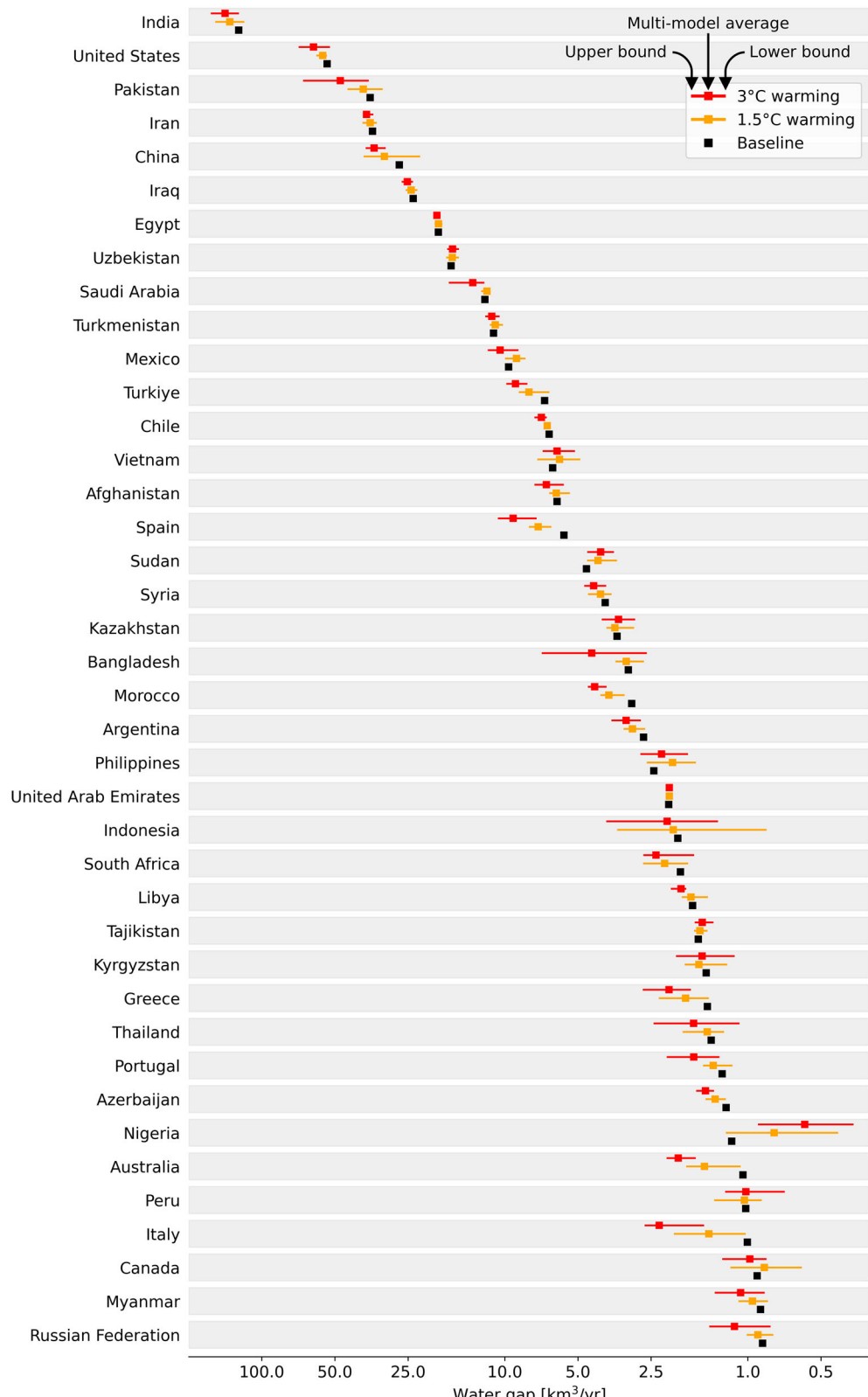

**Fig. 3 | The 40 countries with the largest water gaps worldwide.** Water gaps are presented under baseline (black), 1.5 °C (yellow), and 3 °C (red) warmer climate. Interval bars indicate the variability between the five climate models considered in this study. The countries are listed from the largest to the smallest water gaps under baseline climate conditions. X axis data are presented using log10 scale. Detailed model-specific and scenario-specific results are presented in Supplementary Table 1.

to increase by 5.8% (26.5 km³/yr) under a 1.5 °C warming scenario and 14.7% (67.4 km³/yr) under a 3 °C scenario, with the impacts intensifying as temperatures rise (see Table 1 for model-specific- and scenario-specific results). The effects of warming between 1.5 °C and 3 °C are

uneven, with the higher temperature scenario exacerbating issues like groundwater depletion, ecological stress, and unsustainable water use more severely. Even relatively modest increases in the water gap can intensify these problems, putting additional pressure on ecosystems

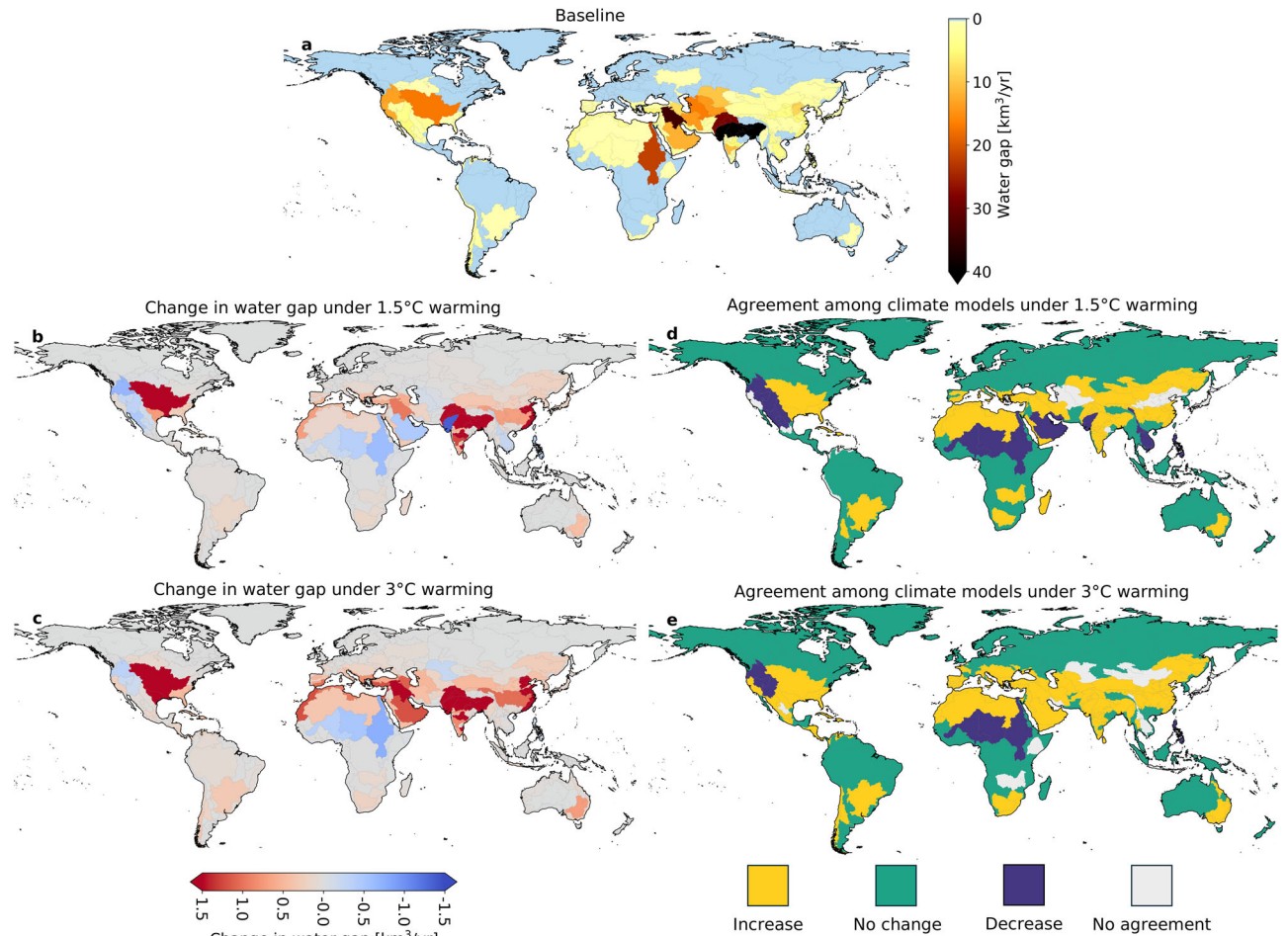

**Fig. 4 | Water gaps in major hydrological basins of the world. a** Water gaps for the baseline period (2001–2010), and **b**, **c** variations in water gaps in a 1.5 °C and 3 °C warmer climate. The maps show the multi-model average among the five climate models considered in this study. Agreement among climate models regarding trends in water gap variations in a 1.5 °C (**d**) and 3 °C (**e**) warmer climate.

Exacerbation of water gaps is shown in yellow, no change in green, and a decrease in water gaps in blue. Gray basins indicate a lack of agreement among climate models. The outline of the major hydrological basins of the world is based on data from the Food and Agriculture Organization of the United Nations[75].

and leading to long-term degradation and difficulty in maintaining freshwater resources.

While these projected increases are concerning, they may still be mitigated through improved water management practices and policy interventions. This includes investing in resilient infrastructure, enhancing water storage capabilities[49,50], desalination of seawater[2], reuse of treated wastewater[2,51,52], and physical and virtual water transfers to distribute water from areas of abundance to areas of need[53,54]. By combining these approaches, we can develop a more resilient and sustainable water management system to combat water scarcity.

Increased water gaps in major agricultural regions could lead to more severe water shortages for irrigation, affecting crop yields and food security. This is particularly critical for countries like India, Pakistan, Iran, Iraq and Egypt. Farmers may need to shift to less water-intensive crops[55], improve crop water productivity[56], or invest in more efficient irrigation technologies[57] to adapt to changing water availability. Maintaining environmental flow requirements is crucial for ecosystem health[3]. Increased water gaps could threaten aquatic ecosystems, leading to loss of biodiversity and degradation of natural habitats. There will be a need for intensified conservation efforts to protect and restore affected ecosystems, especially in basins with important projected increases in water gaps.

As economic theories suggest that water supply and demand will eventually balance, the idea of persistent long-term water gaps is

debatable. Mechanisms like virtual water trade—importing water-intensive products from water-rich regions—can mitigate local water stress by redistributing water usage[58,59]. However, such trade alone cannot address the environmental damage caused by unsustainable water use[12,60]. Without strong policies, farmers may overexploit resources, compromising environmental quality and future sustainability[58,59]. While economic alignment is possible, policies and strategic management are essential to prevent ecological degradation and ensure sustainable water use[58,59].

Climate models have limitations in accurately simulating rainfall[61], particularly in monsoon-driven regions, which affects our findings. Three of the top five river basins with significant water gaps—Ganges-Brahmaputra, Indo, and Sabarmati—are in the South Asian Monsoon region, where these limitations are most pronounced (Fig. 5). The 6th IPCC report highlights that climate models struggle to reproduce monsoon behavior and historical rainfall trends, leading to lower confidence in related projections[61]. Our analysis acknowledges this uncertainty (Figs. 3 and 5), especially in monsoon-affected basins, and emphasizes the importance of adaptive water resource management strategies. While addressing these model limitations is outside the scope of our study, we use state-of-the-art climate models that represent current standards in climate assessments. Future improvements in model accuracy, particularly for monsoon dynamics, will be crucial for refining projections and reducing uncertainties.

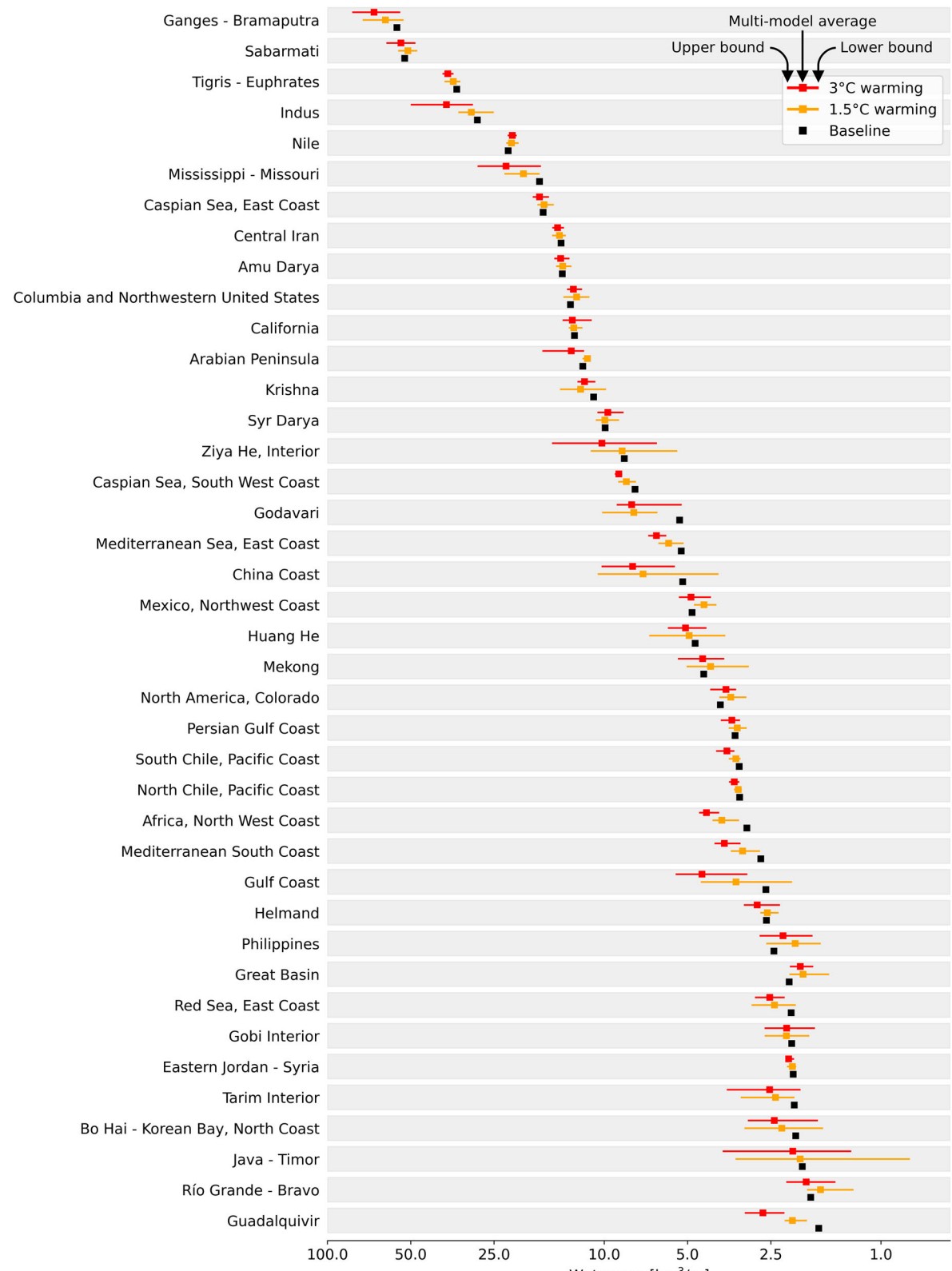

**Fig. 5 | The 40 major hydrological basins with the largest water gaps worldwide.** Water gaps are presented under baseline (black), 1.5 °C (orange), and 3 °C (red) warmer climate. Interval bars indicate the variability between the five climate models considered in this study. The name and outline of major hydrological basins of the world are based on data from the Food and Agriculture Organization of the United Nations[75]. The major hydrological basins are listed from the largest to the smallest water gaps under baseline climate conditions. X-axis data are presented using log10 scale. Detailed model-specific and scenario-specific results are presented in Supplementary Table 2.

Conducting a global analysis involves making appropriate assumptions to manage its complexity. We applied a well-established methodology and input data to evaluate water gaps. Global hydrological models are essential for assessing large-scale trends and aiding water resource planning but face limitations at local levels, particularly in the Global South, due to data calibration that often relies on more robust data from the Global North. This disparity can lead to incomplete representations of local conditions in areas vulnerable to water scarcity and climate impacts. The lack of reliable local data complicates model validation, yet waiting for comprehensive data could delay progress for years. A balanced approach is necessary: using current global assessments with transparent acknowledgment of their limitations while pushing for improved local data collection and validation. Investing in better monitoring in the Global South can progressively enhance the precision and local relevance of future analyses, supporting more effective adaptation strategies.

Direct comparisons with previous studies are challenging due to differences in input data for water consumption and runoff, period of the assessment, the extent of irrigated areas, inter- and intra-annual variability, the time steps of hydrological models (monthly versus annual), and varying assumptions about environmental flow allocations. Nonetheless, we note that our baseline period results are consistent with findings from earlier research. Specifically, our findings align with earlier research that estimated around half of global water consumption was unsustainable for the same period. We estimate unsustainable consumption at 457.9 km$^3$ per year, compared to other studies that have reported values ranging from 408 to 525 km$^3$ per year, depending on assumptions, input data, and environmental flow allocations[1,4,10,12,13]. Country-specific unsustainable consumption is consistent with previous work, which used similar assumptions but different input data[1,4,12].

In the well-studied California Central Valley, groundwater depletion has been estimated and validated at 7–9 km$^3$/year, with total water storage losses ranging from 10–11 km$^3$/year[62]. Other research reports groundwater depletion at 8.58 km$^3$/year for California[63], aligning with our estimate of 12.8 km$^3$/year. Similarly, in the extensively studied Colorado River Basin, we estimate unsustainable water consumption at 3.8 km$^3$/year, consistent with previous estimates of 3–4 km$^3$/year[64].

For Pakistan, previous work reported water gaps of $37 \pm 12$ km$^3$/year[8], respectively, which align with our estimate of 35.8 km$^3$/year. In Egypt, where the water gap has been estimated at ~20 km$^3$/year[65], we estimate it to be 18.8 km$^3$/year. For Iran, Libya, Saudi Arabia, and Mexico, studies have estimated groundwater depletion-related water gaps of 33.3 km$^3$, 2.5 km$^3$, 12.5 km$^3$, and 11.1 km$^3$ per year[60], while our estimates are 35 km$^3$, 1.7 km$^3$, 12.1 km$^3$, and 9.6 km$^3$ per year, respectively.

The variability of water gaps under 1.5 °C and 3 °C warming highlights the importance of continued efforts to mitigate climate change by reducing greenhouse gas emissions to limit its impact on water resources. Our findings emphasize the urgent need for integrated water resource management and robust climate adaptation strategies to address the challenges posed by changing water availability due to global warming.

## Methods
We used climate outputs from five climate models from the CMIP6 archive to quantify water gaps under baseline, 1.5 °C, and 3 °C warming scenarios with respect to the preindustrial era. Water gaps are defined as the difference between renewable water availability and water consumption while maintaining adequate flows in aquatic environments[1]. A water gap arises when human water consumption exceeds the renewable water supply from rivers, lakes, and shallow aquifers replenished by rainfall. Water gaps are assessed by solving a water balance at a 30-arc minute resolution (~50 km at the Equator), utilizing monthly surface and subsurface runoff data from five CMIP6

climate model outputs[47]. Water consumption data was taken from Huang et al.[66]. The baseline scenario refers to the 2001–2010 period. The 1.5 °C scenario represents the global target set in the 2015 Paris Climate Agreement, while the 3 °C scenario represents the level of warming expected by the end of the century if greenhouse gas emissions are not reduced.

### Runoff data
We used monthly runoff (surface and subsurface) under baseline, 1.5 °C, and 3 °C warming scenarios (Supplementary Fig. 6). Local monthly surface and subsurface runoff estimates for the baseline climate were sourced from historical observations from PCR-GLOBWB 2.0[67]. For future climate projections, we utilized CMIP6 SSP5-RCP8.5 data and retrieved monthly runoff data (both surface and subsurface) from five global climate models (IPSL-CM6A-LR, UKESM1-0-LL, GFDL ESM4, MPI-ESM1.2-HR, MRI-ESM2.0) and one global hydrological model (H08)[68] provided by the Inter-Sectoral Impact Model Intercomparison Project 3b (https://www.isimip.org/)[69]. We modeled the 1.5 °C and 3 °C warming scenarios under the CMIP6 SSP5-RCP8.5 pathway to represent climate change under a business-as-usual policy framework. In total, we analyzed 10 climate outputs: 5 for the 1.5 °C scenario and 5 for the 3 °C scenario.

For each climate output, we calculated the difference between projected and historical runoff and incorporated this difference into the baseline observed runoff data. Adding the perturbation (model-specific projection climate minus model-specific historical climate) to observed reference climate data is a standard method in climate model analysis[24,48,70]. This adjustment is necessary due to variations in historical climates among models, limited observations, and differing assumptions about climate forcing terms, such as aerosols and clouds that strongly influence the regional distributions of precipitation, and land use. Each CMIP6 model also handles aerosols and clouds differently, affecting regional precipitation distributions.

The rationale for selecting GFDL-ESM4, IPSL-CM6A-LR, MPI-ESM1-2-HR, MRI-ESM2-0, and UKESM1-0-LL as climate models lies in their structural uniqueness and their collective representation of the broader CMIP6 ensemble. Specifically, GFDL-ESM4, MPI-ESM1-2-HR, and MRI-ESM2-0 exhibit lower climate sensitivity, whereas IPSL-CM6A-LR and UKESM1-0-LL display higher sensitivity[71]. These differing sensitivities result in varied warming trajectories up to the year 2100. Using logistic regression[72], we determined the years when each model is projected to reach the 1.5 °C or 3 °C warming thresholds with a 0.5 probability: 2010 (2042), 2019 (2039), 2035 (2068), 2028 (2064), and 2018 (2048) for IPSL-CM6A-LR, UKESM1-0-LL, GFDL-ESM4, MPI-ESM1-2-HR, and MRI-ESM2-0 at 1.5 °C (3 °C) warming, respectively. Each decade is bracketed by the corresponding midpoint year signifying a 0.5 probability and is accompanied by a ±5-year window (Supplementary Fig. 7). In summary, the climate models estimate that the average global temperature during the baseline period is 0.6 °C warmer than the preindustrial era (1850–1900) and that the 1.5 °C and 3 °C warming scenarios will be reached in 2010–2035 and 2039–2068, respectively (Supplementary Fig. 7).

### Water consumption data
Consumptive water refers to water that is not returned to the environment but is either evaporated or embedded in goods. Baseline monthly water consumption of different sectors—irrigation, livestock, electricity generation, domestic, mining, and manufacturing—are taken from Huang et al.[66]. This dataset provides monthly water consumption figures from four hydrological models (WaterGAP, H08, LPJmL, PCR-GLOBWB) for the baseline period (2001–2010)[66]. This dataset is widely utilized within the field of hydrological modeling. The water consumption data it contains has been validated through observational studies and subjected to uncertainty analyses[66]. For example, country-specific water withdrawal data was validated using

the FAO AQUASTAT database. The validation results demonstrate a good fit with the available data points, including several from Global South[66]. These efforts underscore the reliability and robustness of the dataset for comprehensive global hydrological assessments. Irrigation water consumption data incorporates factors such as seasonality, crop growth calendars, and varying rainfall patterns throughout the year. This approach ensures that the data reflects realistic water demands based on temporal and regional climatic variations.

Future monthly irrigation water consumption data were obtained from CMIP6 SSP5-RCP8.5 outputs from five global climate models (IPSL-CM6A-LR, UKESM1-0-LL, GFDL ESM4, MPI-ESM1.2-HR, MRI-ESM2.0) and one global hydrological model (H08)[68]. Similarly to the future runoff data, we added the difference between historical and future irrigation water consumption to the baseline irrigation consumption. This common method in climate model analysis involves adding the perturbation (the difference between model-specific projected climate and model-specific historical climate) to reference data[24,48,70]. Water consumption for other sectors (livestock, electricity generation, domestic, mining, and manufacturing) was held stable across the warming scenarios. This approach was chosen to avoid introducing additional uncertainty into our results and ensure a clearer assessment of the impact of climate scenarios on water use.

### Limitations of water consumption data resolution
The water consumption data used in this study were validated using hydrological models capable of capturing seasonal variability and human water usage patterns at regional and basin scales[66]. A notable limitation, however, is the absence of high-resolution data at sub-national levels, which can affect the precision of assessments in specific regions, particularly in tropical areas. These areas experience highly dynamic water consumption due to diverse crop calendars and monsoon-driven rainfall patterns. While the models employed offer a robust framework built on the best available datasets, we recognize that local variability, especially in smaller countries and basins, may not be comprehensively represented.

Moreover, the scale of irrigation areas influences water consumption results, and uncertainties regarding the actual size of these areas introduce challenges to accurately estimating irrigation water usage[73]. Addressing these limitations requires integrating more detailed, region-specific data to improve the precision of water consumption estimates.

Future efforts should focus on incorporating higher-resolution monitoring and validation data to better capture water consumption dynamics at finer spatial scales. Enhancing the quality and granularity of data collection will be essential for refining model outputs and supporting more targeted and effective local water management strategies. This study provides a foundation for future regional analyses at higher resolutions, highlighting areas where detailed follow-up studies are most needed and identifying regions less likely to face important water gaps.

### Assessment of renewable water availability
Renewable blue water availability at a 30-arc minute resolution was evaluated as the difference between blue water flows generated within each grid cell and the environmental flow requirements. Renewable blue water availability encompasses surface water, and groundwater volumes recharged through the hydrological cycle[13]. Monthly blue water flows were assessed from local runoff estimates, calculated using an upstream-downstream flow accumulation routing module[1,4,12,24,48]. Blue water flows are supplied to a grid cell either as local runoff (surface and subsurface), and inflow from upstream grid cells along river networks based on an upstream-downstream flow accumulation river routing module. To ensure the health of natural water ecosystems, a portion of blue water flows must be preserved for environmental needs[74]. Environmental flow requirements were determined using the

Variable Monthly Flow method[74], which accounts for the seasonality of blue water flows when estimating environmental flow needs.

### Assessment of water gaps
Water gaps, expressed in volumes in cubic meters, are defined as the difference between available water resources and water consumption in a particular region. For a given climate model, month, and pixel, a water gap exists when renewable water availability is less than water consumption. Water gaps at the major hydrological basin- and country scale were then assessed by summing the water gaps in each pixel within the country and basin.

## Data availability
Source data are provided with this paper. Additional data and results from this study are available at: https://doi.org/10.5281/zenodo.13220763.

## Code availability
Data analysis was performed using Python 3.11.0, utilizing the following packages: PySheds (0.3.5), Xarray (2022.11.0), Cartopy (0.21.1), GeoPandas (0.12.1), Matplotlib (3.6.2), Pandas (1.5.2), and NumPy (1.23.5). Scripts and functions developed for this study are available upon reasonable request from the corresponding author.

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

## Acknowledgements

This research received support through Schmidt Sciences, LLC.

## Author contributions

L.R. and M.S. conceived the study, performed analysis, analyzed data, and wrote the study.

## Competing interests

The authors declare no competing interests.
