## [Transparent Peer Review file · Nature Communications]

Global water gaps under future warming levels

Corresponding Author: Dr Lorenzo Rosa

Version 0:

Reviewer comments:

Reviewer #1

(Remarks to the Author)

This is a highly interesting and timely manuscript, and I commend the authors for tackling such challenging scientific questions with important policy implications. While global assessments like this are essential, I have serious concerns about their applicability at local scales. This stems primarily from the fact that much of the WaterGAP water use data is calibrated based on regions in the Global North, whereas the major adaptation gaps exist in the Global South. Extensive validation of water use data at a local scale in Global South countries is urgently needed, though data availability remains a significant challenge.

The critical question is: Should we halt progress due to the lack of data, or should we focus on extensive monitoring to enable more robust analysis, even though this could take a decade? This debate will persist, and I believe it should be flagged as a caveat in the manuscript.

My specific comments are as follows:

Water use data exhibits significant variability within countries, across basins, and between seasons, especially in tropical regions where it fluctuates with crop calendars and monsoons. The human component is often not accurately represented in estimating water demand and use. I recommend the authors include a thorough validation of water use, particularly for the river basins highlighted in the results.

Climate models have known limitations in simulating rainfall, particularly in monsoon regions. In Figure 5, three out of the top five river basins are located in the South Asian Monsoon region. The IPCC AR6 WG1 report states that climate models have low confidence in simulating monsoon behavior, largely because they fail to reproduce historical trends. What are the implications of these limitations for the final results?

I would also like to see a more extensive validation, at least for the historical period simulations. While I'm unsure about data availability, the authors could perform a comprehensive literature review at the river basin level to assess whether their historical results align with those reported in the literature.

In conclusion, while I appreciate the authors' efforts, my primary concern remains with validation. I am confident that the authors can address these issues to produce an even more impactful article.

Reviewer #2

(Remarks to the Author)

Global water gaps under future warming levels assess the future impacts of climate change on global water resources, expressed in terms of water 'gaps'. Globally, an increase of 5% (13%) is found for the 1.5 degree (3 degree) warming scenario, with regional variation.

I am not convinced by the suggestion/claim of the authors that they introduce <for the first time> the concept of water 'gaps' or that it is indeed something novel. Water gap analysis has been around since at least the Charting our water future report of McKinsey and the World Bank in 2009 and applied by many since. The way the authors define it, it is basically unsustainable water use. I fail to see why indicators have so far failed to quantify volumetric water deficits. (L58) Don't we have unsustainable (ground)water use studies in volumetric terms (e.g. the PCR-GLOBWB (ground)model studies), and

models assessing exceedance of planetary boundaries? (Gerten et al, using LPJmL).

A criticism of future water 'gaps' has been that from an economic perspective, in the long run, demand will (have to) meet supply. There might be temporary imbalances, but the emergence and/or continuation over decades of water 'gaps' could be questioned. The authors do not address this, or only as a footnote in the discussion. Will other developments, such as shifting trade patterns, not reduce these 'gaps'? Secondly, how robust are these results, the 'gaps', if the supply side is based on a different set of models and climate scenarios than the demand (consumption) side? And are future consumption levels determined separately for +1.5 and +3 degree – it does not become clear.

More in general, the methods applied, and data used are fairly standard. The models from which the consumption data was taken (WaterGAP, H08, LPJmL, PCR-GLOBWB) all have been used for similar global or regional analyses. A comparison of results could be interesting. The paper does not add much, e.g. by elaborating what sectors contribute most, or what drivers are most important. Variability between years is not explicitly addressed. Agriculture will remain the largest water consumer but nowhere in the paper are food or crops even mentioned.

Finally, the study does not present any new or remarkable results. Water supply and demand changes in the most important regions/basins have been studied in much more detail by others; e.g. on Ganges-Brahmaputra, by Lutz et al, Biemans et al, Wijngaard et al, or for major groundwater depletion regions, by Wada And Bierkens et al.. Increases in water gap of 5% (+1.5) up to 13% in the +3 degree scenario do not appear to be of a magnitude that can't be dealt with through improved practices.

Version 1:

Reviewer comments:

Reviewer #1

(Remarks to the Author)

The authors have addressed all my comments. I congratulate them for an excellent work.

Reviewer #2

(Remarks to the Author)

The authors have addressed most of my concerns in detail. I have two further comments:

1. the authors might want to sharpen the novelty of their study still further. I would think most/many research nowadays incorporates future (climate) projections (contrary to what is claimed in L87). And if past research has focused on 'quantifying groundwater depletion' and 'environmental flow reduction' (L83-84) then it has addressed water gaps as defined by the authors, no? That would leave the (global) multi-model approach.

2. I do believe the results could still be interpreted better, with a discussion of what a 5% or 13% increase in 'water gaps' means, globally, compared to the present-day situation in which there already exist significant water 'gaps'. I do not want to downplay the potential consequences of increases of 5-13% (though with hardly any change in 8 out of 11 largest basins, figure 5), but I wonder if, rather than (only) focusing on the change and the need for comprehensive climate adaptation (as in "... highlighting substantial challenges for water sustainability" (L237) and "These results highlight the need for comprehensive climate adaptation plans that account for changing water availability."(L244)), the study shows that dealing with existing water gaps is important. This would probably also help adapting to future change in most basins.

Also, interestingly, most increase and higher uncertainty occurs in the smaller 20 of basins (figure 5). Why is that? Do changes simply average out in the larger basins? Could that mean that within those larger basins, there could still be considerable changes in supply and demand?

Version 2:

Reviewer comments:

Reviewer #2

(Remarks to the Author)

The authors have addressed all my comments.

Reviewer #1

This is a highly interesting and timely manuscript, and I commend the authors for tackling such challenging scientific questions with important policy implications.

We appreciate the reviewer’s positive feedback and constructive comments. Below, we addressed all points raised as requested.

While global assessments like this are essential, I have serious concerns about their applicability at local scales. This stems primarily from the fact that much of the WaterGAP water use data is calibrated based on regions in the Global North, whereas the major adaptation gaps exist in the Global South. Extensive validation of water use data at a local scale in Global South countries is urgently needed, though data availability remains a significant challenge.

The critical question is: Should we halt progress due to the lack of data, or should we focus on extensive monitoring to enable more robust analysis, even though this could take a decade? This debate will persist, and I believe it should be flagged as a caveat in the manuscript.

Thank you for pointing out this important point. We agree with the reviewer’s comment, and we flagged this caveat in the revised manuscript, which reads:

“Conducting a global analysis involves making appropriate assumptions to manage its complexity. We applied a well-established methodology and input data to evaluate water gaps, defined as unsustainable water consumption, under both baseline conditions and future warming scenarios. Global hydrological models are essential for assessing large-scale trends and aiding water resource planning but face limitations at local levels, particularly in the Global South, due to data calibration that often relies on more robust data from the Global North. This disparity can lead to incomplete representations of local conditions in areas vulnerable to water scarcity and climate impacts. The lack of reliable local data complicates model validation, yet waiting for comprehensive data could delay progress for years. A balanced approach is necessary: using current global assessments with transparent acknowledgment of their limitations while pushing for improved local data collection and validation. Investing in better monitoring in the Global South can progressively enhance the precision and local relevance of future analyses, supporting more effective adaptation strategies.”

The methods section reads:

“Baseline monthly water consumption of different sectors — irrigation, livestock, electricity generation, domestic, mining, and manufacturing — are taken from Huang et al., 2018. This dataset provides monthly water consumption figures from four hydrological models (WaterGAP, H08, LPJmL, PCR-GLOBWB) for the baseline period (2001-2010). This dataset is widely utilized within the field of hydrological modeling. The water consumption data it contains has been validated through observational studies and subjected to uncertainty analyses (Huang et al., 2018). Country-specific water withdrawal data was validated using the FAO AQUASTAT database. The validation results demonstrate a good fit with the available data points, including

several from Global South (Huang et al., 2018). These efforts underscore the reliability and robustness of the dataset for comprehensive global hydrological assessments. Irrigation water consumption data incorporates factors such as seasonality, crop growth calendars, and varying rainfall patterns throughout the year. This approach ensures that the data reflects realistic water demands based on temporal and regional climatic variations.”

References:

Huang, Z., Hejazi, M., Li, X., Tang, Q., Vernon, C., Leng, G., ... & Wada, Y. (2018). Reconstruction of global gridded monthly sectoral water withdrawals for 1971–2010 and analysis of their spatiotemporal patterns. *Hydrology and Earth System Sciences*, 22(4), 2117-2133.

My specific comments are as follows:

Water use data exhibits significant variability within countries, across basins, and between seasons, especially in tropical regions where it fluctuates with crop calendars and monsoons. The human component is often not accurately represented in estimating water demand and use. I recommend the authors include a thorough validation of water use, particularly for the river basins highlighted in the results.

Thank you for pointing this out.

We used water consumption data from a well-established open access multi-model dataset, which has been validated through observational studies and subjected to uncertainty analyses (Huang et al., 2018). Irrigation water consumption data are derived from crop water models that account for seasonality, crop calendars and different rainfall regimes across the year.

The methods section now reads:

“Consumptive water refers to water that is not returned to the environment but is either evaporated or embedded in goods⁶³. Baseline monthly water consumption of different sectors — irrigation, livestock, electricity generation, domestic, mining, and manufacturing — are taken from Huang et al., 2018. This dataset provides monthly water consumption figures from four hydrological models (WaterGAP, H08, LPJmL, PCR-GLOBWB) for the baseline period (2001-2010). This dataset is widely utilized within the field of hydrological modeling. The water consumption data it contains has been validated through observational studies and subjected to uncertainty analyses (Huang et al., 2018). Country-specific water withdrawal data was validated using the FAO AQUASTAT database. The validation results demonstrate a good fit with the available data points, including several from Global South (Huang et al., 2018). These efforts underscore the reliability and robustness of the dataset for comprehensive global hydrological assessments. Irrigation water consumption data incorporates factors such as seasonality, crop growth calendars, and varying rainfall patterns throughout the year. This approach ensures

that the data reflects realistic water demands based on temporal and regional climatic variations.”

Reference:

Huang, Z., Hejazi, M., Li, X., Tang, Q., Vernon, C., Leng, G., ... & Wada, Y. (2018). Reconstruction of global gridded monthly sectoral water withdrawals for 1971–2010 and analysis of their spatiotemporal patterns. *Hydrology and Earth System Sciences*, 22(4), 2117-2133.

We also acknowledge this important point in the revised text, which reads:

“Limitations of Water Consumption Data Resolution

The water consumption data used in this study were validated using hydrological models capable of capturing seasonal variability and human water usage patterns at regional and basin scales (Huang et al., 2018). A notable limitation, however, is the absence of high-resolution data at sub-national levels, which can affect the precision of assessments in specific regions, particularly in tropical areas. These areas experience highly dynamic water consumption due to diverse crop calendars and monsoon-driven rainfall patterns. While the models employed offer a robust framework built on the best available datasets, we recognize that local variability, especially in smaller countries and basins, may not be comprehensively represented.

Moreover, the scale of irrigation areas significantly influences water consumption results, and uncertainties regarding the actual size of these areas introduce challenges to accurately estimating irrigation water usage (Puy et al., 2021). Addressing these limitations requires integrating more detailed, region-specific data to improve the precision of water consumption estimates.

Future efforts should focus on incorporating higher-resolution monitoring and validation data to better capture water consumption dynamics at finer spatial scales. Enhancing the quality and granularity of data collection will be essential for refining model outputs and supporting more targeted and effective local water management strategies. This study provides a foundation for future regional analyses at higher resolutions, highlighting areas where detailed follow-up studies are most needed and identifying regions less likely to face significant water gaps.”

Reference:

Huang, Z., Hejazi, M., Li, X., Tang, Q., Vernon, C., Leng, G., ... & Wada, Y. (2018). Reconstruction of global gridded monthly sectoral water withdrawals for 1971–2010 and analysis of their spatiotemporal patterns. *Hydrology and Earth System Sciences*, 22(4), 2117-2133.

Puy, A., Borgonovo, E., Lo Piano, S., Levin, S.A. and Saltelli, A., 2021. Irrigated areas drive irrigation water withdrawals. *Nature communications*, 12(1), p.4525.

Climate models have known limitations in simulating rainfall, particularly in monsoon regions. In Figure 5, three out of the top five river basins are located in the South Asian Monsoon region. The IPCC AR6 WG1 report states that climate models have low confidence in simulating monsoon behavior, largely because they fail to reproduce historical trends. What are the implications of these limitations for the final results?

We acknowledge this important point in the revised text, the discussion section reads:

“Climate models have limitations in accurately simulating rainfall (IPCC, 2021), particularly in monsoon-driven regions, which affects our findings. Three of the top five river basins with significant water gaps—Ganges-Brahmaputra, Indo, and Sabarmati—are in the South Asian Monsoon region, where these limitations are most pronounced (Figure 5). The IPCC AR6 report highlights that climate models struggle to reproduce monsoon behavior and historical rainfall trends, leading to lower confidence in related projections (IPCC, 2021). Our analysis acknowledges this uncertainty, especially in monsoon-affected basins, and emphasizes the importance of adaptive water resource management strategies. While addressing these model limitations is outside the scope of our study, we use state-of-the-art climate models that represent current standards in climate assessments. Future improvements in model accuracy, particularly for monsoon dynamics, will be crucial for refining projections and reducing uncertainties.”

References

IPCC, 2021: Climate Change 2021: The Physical Science Basis. Contribution of Working Group I to the Sixth Assessment Report of the Intergovernmental Panel on Climate Change[Masson-Delmotte, V., P. Zhai, A. Pirani, S.L. Connors, C. Péan, S. Berger, N. Caud, Y. Chen, L. Goldfarb, M.I. Gomis, M. Huang, K. Leitzell, E. Lonnoy, J.B.R. Matthews, T.K. Maycock, T. Waterfield, O. Yelekçi, R. Yu, and B. Zhou (eds.)]. Cambridge University Press, Cambridge, United Kingdom and New York, NY, USA, In press, doi:[10.1017/9781009157896](https://doi.org/10.1017/9781009157896).

I would also like to see a more extensive validation, at least for the historical period simulations. While I’m unsure about data availability, the authors could perform a comprehensive literature review at the river basin level to assess whether their historical results align with those reported in the literature. In conclusion, while I appreciate the authors’ efforts, my primary concern remains with validation. I am confident that the authors can address these issues to produce an even more impactful article.

We thank the reviewer for the constructive review. We have addressed the reviewer’s comment by comparing our results with previous work that assessed past groundwater depletion or unsustainable water consumption. The discussion section now reads:

“Direct comparisons with previous studies are challenging due to differences in input data for water consumption and runoff, time of the assessment, the extent of irrigated areas, inter- and intra-annual variability, the time-steps of hydrological models (monthly versus annual), and varying assumptions about environmental flow allocations. Nonetheless, we note that our baseline period results are consistent with findings from earlier research. Specifically, our findings align with earlier research that estimated around half of global water consumption was

unsustainable for the same period. We estimate unsustainable consumption at 456.7 km³ per year, compared to other studies that have reported values ranging from 408 to 525 km³ per year, depending on assumptions, input data, and environmental flow allocations (Mekonnen and Hoekstra, 2016, 2020; Brauman et al., 2016; Jägermeyr et al., 2017; Rosa et al., 2018, 2019, 2020; Mekonnen et al., 2024). Country-specific unsustainable consumption is consistent with previous work, which used similar assumptions but different input data (Rosa et al., 2018; 2019; 2020; Mekonnen and Hoekstra, 2020).

In the well-studied California Central Valley, groundwater depletion has been estimated and validated at 7–9 km³/year, with total water storage losses ranging from 10–11 km³/year (Scanlon et al., 2012). Other research reports groundwater depletion at 8.58 km³/year for California (Liu et al., 2022), aligning with our estimate of 12.8 km³/year. Similarly, in the extensively studied Colorado River Basin, we estimate unsustainable water consumption at 3.8 km³/year, consistent with previous estimates of 3–4 km³/year (Richter et al., 2023).

For Pakistan and Algeria, previous work reported water gaps of 37 ± 12 km³/year and 1.7 ± 0.6 km³/year, respectively (Wada et al., 2010), which align closely with our estimates of 35.8 km³/year and 0.8 km³/year. In Egypt, where the water gap has been estimated at around 20 km³/year (Nikiel and Eltahir, 2021), we estimate it to be 18.8 km³/year. For Iran, Saudi Arabia, Mexico, and Libya studies have estimated groundwater depletion-related water gaps of 33.3 km³, 12.5 km³, 11.1 km³, and 2.5 km³ per year (Dalin et al., 2017), while our estimates are 35 km³, 12.1 km³, 9.6 km³, and 1.7 km³ per year, respectively.”

References

Dalin, C., Wada, Y., Kastner, T. and Puma, M.J., 2017. Groundwater depletion embedded in international food trade. *Nature*, 543(7647), pp.700-704.

Jägermeyr, J., Pastor, A., Biemans, H. and Gerten, D., 2017. Reconciling irrigated food production with environmental flows for Sustainable Development Goals implementation. *Nature communications*, 8(1), p.15900.

Liu, P.W., Famiglietti, J.S., Purdy, A.J., Adams, K.H., McEvoy, A.L., Reager, J.T., Bindlish, R., Wiese, D.N., David, C.H. and Rodell, M., 2022. Groundwater depletion in California’s Central Valley accelerates during megadrought. *Nature Communications*, 13(1), p.7825.

Mekonnen, M.M. and Hoekstra, A.Y., 2016. Four billion people facing severe water scarcity. *Science advances*, 2(2), p.e1500323.

Mekonnen, M.M. and Hoekstra, A.Y., 2020. Blue water footprint linked to national consumption and international trade is unsustainable. *Nature Food*, 1(12), pp.792-800.

Nikiel, C.A. and Eltahir, E.A., 2021. Past and future trends of Egypt’s water consumption and its sources. *Nature Communications*, 12(1), p.4508.

Richter, B.D., Lamsal, G., Marston, L., Dhakal, S., Sangha, L.S., Rushforth, R.R., Wei, D., Ruddell, B.L., Davis, K.F., Hernandez-Cruz, A. and Sandoval-Solis, S., 2024. New water

accounting reveals why the Colorado River no longer reaches the sea. *Communications Earth & Environment*, 5(1), p.134.

Rosa, L., Chiarelli, D.D., Rulli, M.C., Dell'Angelo, J. and D'Odorico, P., 2020. Global agricultural economic water scarcity. *Science Advances*, 6(18), p.eaaz6031.

Rosa, L., Chiarelli, D.D., Tu, C., Rulli, M.C. and D'Odorico, P., 2019. Global unsustainable virtual water flows in agricultural trade. *Environmental Research Letters*, 14(11), p.114001.

Rosa, L., Rulli, M.C., Davis, K.F., Chiarelli, D.D., Passera, C. and D'Odorico, P., 2018. Closing the yield gap while ensuring water sustainability. *Environmental Research Letters*, 13(10), p.104002.

Scanlon, B.R., Longuevergne, L. and Long, D., 2012. Ground referencing GRACE satellite estimates of groundwater storage changes in the California Central Valley, USA. *Water Resources Research*, 48(4).

Wada, Y., van Beek, L.P. and Bierkens, M.F., 2012. Nonsustainable groundwater sustaining irrigation: A global assessment. *Water Resources Research*, 48(6).

Reviewer #2

Global water gaps under future warming levels assess the future impacts of climate change on global water resources, expressed in terms of water 'gaps'. Globally, an increase of 5% (13%) is found for the 1.5 degree (3 degree) warming scenario, with regional variation.

We appreciate the reviewer's critical and constructive comments. Below, we addressed all points raised as requested.

I am not convinced by the suggestion/claim of the authors that they introduce <for the first time> the concept of water 'gaps' or that it is indeed something novel. Water gap analysis has been around since at least the Charting our water future report of McKinsey and the World Bank in 2009 and applied by many since. The way the authors define it, it is basically unsustainable water use.

I fail to see why indicators have so far failed to quantify volumetric water deficits. (L58) Don't we have unsustainable (ground)water use studies in volumetric terms (e.g. the PCR-GLOBWB (ground)model studies), and models assessing exceedance of planetary boundaries? (Gerten et al, using LPJmL).

We thank you for pointing this out. We acknowledge that the water gap concept was not properly presented in the text, and we are not the first ones to use it.

In the revised manuscript we have better explained the novelty and importance of our study. As also suggested by reviewer #1, this is a highly interesting and timely manuscript, tackling a challenging scientific question with important policy implications. Previous work mainly quantified unsustainable groundwater use; the novelty of our work is:

- 1) We quantify unsustainable water consumption considering groundwater, rivers, lakes and environmental flows depletion, thereby quantifying water needs for humans and ecosystems.
- 2) We quantify future water gaps with a multi-model analysis showing variability and agreement among hydrological model and climate models and uncertainty in projections.
- 3) We quantify future water gaps considering two climate scenarios

The revised text better explains the novelty, importance and innovation of our work, which reads:

“First introduced in 2009 (Addams et al., 2009) and widely utilized since, water gaps indicate the difference between renewable water availability and water consumption within a specific region. A water gap occurs when, in any given month, water consumption surpasses available renewable water resources. These gaps highlight unsustainable water use, leading to the depletion of groundwater, rivers, lakes, and environmental flow reserves. Such situations signal critical shortages where water resources cannot meet the demands of populations, agriculture, industry, energy production, and ecosystems, potentially resulting in severe social, economic, and environmental consequences. Quantifying water gaps provides decision-makers with vital information for effective water scarcity management.

Past research has primarily concentrated on quantifying groundwater depletion (Konikow and Kendy, 2005; Wada et al., 2010; Gleeson et al., 2012; Robell et al., 2018; Scanlon et al., 2023) and environmental flow reduction (Gerten et al., 2013; Jägermeyer et al., 2017; Rosa et al., 2019; Pastor et al., 2019; de Graaf et al., 2019; Gerten et al., 2020) on a global scale. Other studies have explored groundwater depletion and unsustainable water use at regional levels (Rodell et al., 2008; Scanlon et al., 2012; Wijngaard et al., 2018; Biemans et al., 2019; Rateb et al., 2020; Lutz et al., 2022). However, comprehensive global assessments using a multi-model approach to evaluate both historical and future water gaps remain lacking. Additionally, most existing research does not incorporate future projections, multi-model and multi-scenario analyses, or sensitivity tests—elements that are essential for informed decision-making and effective adaptation to water scarcity amid uncertainty. This gap underscores the need for comprehensive research that can guide policy and bolster resilience in the context of climate variability and growing water demand.

Here, we quantify baseline and future water gaps using a multi-model analysis that considers two plausible future warming scenarios to assess the uncertainty of projections.....”

“Our study uses publicly available, widely adopted, and validated climate and hydrological model outputs to quantify water gaps under baseline and future warming scenarios. The novelty of our approach lies in its multi-model analysis, which accounts for groundwater depletion and the depletion of surface water and environmental flows, thus estimating water needs for ecosystems. By assessing water deficits during the baseline period (2000–2010) and under two future warming levels, our work provides valuable insights for water resource planning and

adaptation strategies. It identifies varying adaptation needs, highlights key trends and regional vulnerabilities, and pinpoints countries and river basins where water gaps are projected to change substantially, supporting targeted adaptation measures.”

The 2009 McKinsey report (Addams et al., 2009):

<https://www.mckinsey.com/capabilities/sustainability/our-insights/charting-our-water-future>) is a comprehensive 200-page corporate document that evaluates the global water gap and presents cost curves for narrowing this gap. Unlike our study, the report does not incorporate climate model analyses, their variability and uncertainty, and its future projections are limited to 2030. While it employs the concept of a ‘water gap’, its methodology differs significantly from ours, as it does not utilize hydrological or climate models. It is important to note that this report was published in 2009, and computational tools and modeling capabilities have significantly advanced since then.

References:

Addams, L., Boccaletti, G., Kerlin, M. and Stuchtey, M., 2009. Charting our water future. *Economic frameworks to inform decision making, 2030*. McKinsey and Company. <https://www.mckinsey.com/capabilities/sustainability/our-insights/charting-our-water-future>

Biemans, H., Siderius, C., Lutz, A.F., Nepal, S., Ahmad, B., Hassan, T., von Bloh, W., Wijnngaard, R.R., Wester, P., Shrestha, A.B. and Immerzeel, W.W., 2019. Importance of snow and glacier meltwater for agriculture on the Indo-Gangetic Plain. *Nature Sustainability*, 2(7), pp.594-601.

Dalin, C., Wada, Y., Kastner, T. and Puma, M.J., 2017. Groundwater depletion embedded in international food trade. *Nature*, 543(7647), pp.700-704.

de Graaf, I.E., Gleeson, T., Van Beek, L.P.H., Sutanudjaja, E.H. and Bierkens, M.F., 2019. Environmental flow limits to global groundwater pumping. *Nature*, 574(7776), pp.90-94.

Gerten, D., Heck, V., Jägermeyr, J., Boudirsky, B.L., Fetzer, I., Jalava, M., Kummu, M., Lucht, W., Rockström, J., Schaphoff, S. and Schellnhuber, H.J., 2020. Feeding ten billion people is possible within four terrestrial planetary boundaries. *Nature Sustainability*, 3(3), pp.200-208.

Gerten, D., Hoff, H., Rockström, J., Jägermeyr, J., Kummu, M. and Pastor, A.V., 2013. Towards a revised planetary boundary for consumptive freshwater use: role of environmental flow requirements. *Current Opinion in Environmental Sustainability*, 5(6), pp.551-558.

Gleeson, T., Wada, Y., Bierkens, M.F. and Van Beek, L.P., 2012. Water balance of global aquifers revealed by groundwater footprint. *Nature*, 488(7410), pp.197-200.

Jägermeyr, J., Pastor, A., Biemans, H. and Gerten, D., 2017. Reconciling irrigated food production with environmental flows for Sustainable Development Goals implementation. *Nature communications*, 8(1), p.15900.

Konikow, L.F. and Kendy, E., 2005. Groundwater depletion: A global problem. *Hydrogeology journal*, 13, pp.317-320.

Lutz, A.F., Immerzeel, W.W., Siderius, C., Wijngaard, R.R., Nepal, S., Shrestha, A.B., Wester, P. and Biemans, H., 2022. South Asian agriculture increasingly dependent on meltwater and groundwater. *Nature Climate Change*, 12(6), pp.566-573.

Pastor, A.V., Palazzo, A., Havlik, P., Biemans, H., Wada, Y., Obersteiner, M., Kabat, P. and Ludwig, F., 2019. The global nexus of food–trade–water sustaining environmental flows by 2050. *Nature Sustainability*, 2(6), pp.499-507.

Rateb, A., Scanlon, B.R., Pool, D.R., Sun, A., Zhang, Z., Chen, J., Clark, B., Faunt, C.C., Haugh, C.J., Hill, M. and Hobza, C., 2020. Comparison of groundwater storage changes from GRACE satellites with monitoring and modeling of major US aquifers. *Water Resources Research*, 56(12), p.e2020WR027556.

Rodell, M., Velicogna, I. and Famiglietti, J.S., 2009. Satellite-based estimates of groundwater depletion in India. *Nature*, 460(7258), pp.999-1002.

Rosa, L., Chiarelli, D.D., Tu, C., Rulli, M.C. and D’Odorico, P., 2019. Global unsustainable virtual water flows in agricultural trade. *Environmental Research Letters*, 14(11), p.114001.

Scanlon, B.R., Fakhreddine, S., Rateb, A., de Graaf, I., Famiglietti, J., Gleeson, T., Grafton, R.Q., Jobbagy, E., Kebede, S., Kolusu, S.R. and Konikow, L.F., 2023. Global water resources and the role of groundwater in a resilient water future. *Nature Reviews Earth & Environment*, 4(2), pp.87-101.

Scanlon, B.R., Faunt, C.C., Longuevergne, L., Reedy, R.C., Alley, W.M., McGuire, V.L. and McMahon, P.B., 2012. Groundwater depletion and sustainability of irrigation in the US High Plains and Central Valley. *Proceedings of the national academy of sciences*, 109(24), pp.9320-9325.

Wada, Y., Van Beek, L.P., Van Kempen, C.M., Reckman, J.W., Vasak, S. and Bierkens, M.F., 2010. Global depletion of groundwater resources. *Geophysical research letters*, 37(20).

Wijngaard, R.R., Biemans, H., Lutz, A.F., Shrestha, A.B., Wester, P. and Immerzeel, W.W., 2018. Climate change vs. socio-economic development: understanding the future South Asian water gap. *Hydrology and Earth System Sciences*, 22(12), pp.6297-6321.

A criticism of future water ‘gaps’ has been that from an economic perspective, in the long run, demand will (have to) meet supply. There might be temporary imbalances, but the emergence and/or continuation over decades of water ‘gaps’ could be questioned. The authors do not address this, or only as a footnote in the discussion. Will other developments, such as shifting trade patterns, not reduce these ‘gaps’?

Thank you for pointing this out, we have added this important point to the discussion section, which reads:

“As economic theories suggest that water supply and demand will eventually balance, the idea of persistent long-term water gaps is debatable. Mechanisms like virtual water trade—importing water-intensive products from water-rich regions—can mitigate local water stress by redistributing water usage (D’Odorico et al., 2019; Mekonnen et al., 2024). However, such trade alone cannot address the environmental damage caused by unsustainable water use (Dalin et al., 2017; Rosa et al., 2019). Without strong policies, farmers may overexploit resources, compromising environmental quality and future sustainability. While economic alignment is possible, policies and strategic management are essential to prevent ecological degradation and ensure sustainable water use.”

References:

Mekonnen, M.M., Kebede, M.M., Demeke, B.W., Carr, J.A., Chapagain, A., Dalin, C., Debaere, P., D’Odorico, P., Marston, L., Ray, C. and Rosa, L., 2024. Trends and environmental impacts of virtual water trade. *Nature Reviews Earth & Environment*, pp.1-16.

D’Odorico, P., Carr, J., Dalin, C., Dell’Angelo, J., Konar, M., Laio, F., Ridolfi, L., Rosa, L., Suweis, S., Tamea, S. and Tuninetti, M., 2019. Global virtual water trade and the hydrological cycle: patterns, drivers, and socio-environmental impacts. *Environmental Research Letters*, 14(5), p.053001.

Secondly, how robust are these results, the ‘gaps’, if the supply side is based on a different set of models and climate scenarios than the demand (consumption) side? And are future consumption levels determined separately for +1.5 and +3 degree – it does not become clear.

Future water consumption is determined separately for the +1.5°C and +3°C warming scenarios (see Methods). Specifically, irrigation water consumption varies according to the level of warming, while all other types of water consumption are held constant. This approach was chosen to avoid introducing additional uncertainty into our results and ensure a clearer assessment of the impact of climate scenarios on water consumption.

A common method in climate model analysis involves adding the perturbation (the difference between model-specific projected climate and model-specific historical climate) to observed reference climate data. This adjustment is necessary due to variations in historical climates among models, limited observations, and differing assumptions about climate forcing terms, such as aerosols and clouds that strongly influence the regional distributions of precipitation, and land use. Each CMIP6 model also handles aerosols and clouds differently, affecting regional precipitation distributions.

The text reads:

“Future monthly irrigation water consumption data were obtained from CMIP6 RCP8.5 outputs from five global climate models (IPSL-CM6A-LR, UKESM1-0-LL, GFDL ESM4, MPI-

ESM1.2-HR, MRI-ESM2.0) and one global hydrological model (H08)⁵⁸. Similarly to the future runoff data, we added the difference between historical and future irrigation water consumption to the baseline irrigation consumption. This common method in climate model analysis involves adding the perturbation (the difference between model-specific projected climate and model-specific historical climate) to reference data. Water consumption for other sectors (livestock, electricity generation, domestic, mining, and manufacturing) was held constant across the warming scenarios.”

More in general, the methods applied, and data used are fairly standard. The models from which the consumption data was taken (WaterGAP, H08, LPJmL, PCR-GLOBWB) all have been used for similar global or regional analyses. A comparison of results could be interesting. The paper does not add much, e.g. by elaborating what sectors contribute most, or what drivers are most important. Variability between years is not explicitly addressed. Agriculture will remain the largest water consumer but nowhere in the paper are food or crops even mentioned.

We use state-of-the-art methods and data, which are open access. Regarding novelty, please see our previous answer where we better explain the novelty and importance of our work.

The importance of agriculture and irrigation is not better presented in the introduction, results and discussion sections.

We have addressed the reviewer’s comment by comparing our results with previous work. The discussion section now reads:

“Direct comparisons with previous studies are challenging due to differences in input data for water consumption and runoff, time period of the assessment, the extent of irrigated areas, inter- and intra-annual variability, the time-steps of hydrological models (monthly versus annual), and varying assumptions about environmental flow allocations. Nonetheless, we note that our baseline period results are consistent with findings from earlier research. Specifically, our findings align with earlier research that estimated around half of global water consumption was unsustainable for the same period. We estimate unsustainable consumption at 456.7 km³ per year, compared to other studies that have reported values ranging from 408 to 525 km³ per year, depending on assumptions, input data, and environmental flow allocations (Mekonnen and Hoekstra, 2016, 2020; Brauman et al., 2016; Jägermeyr et al., 2017; Rosa et al., 2018, 2019, 2020; Mekonnen et al., 2024). Country-specific unsustainable consumption is consistent with previous work, which used similar assumptions but different input data (Rosa et al., 2018; 2019; 2020; Mekonnen and Hoekstra, 2020).

In the well-studied California Central Valley, groundwater depletion has been estimated and validated at 7–9 km³/year, with total water storage losses ranging from 10–11 km³/year (Scanlon et al., 2012). Other research reports groundwater depletion at 8.58 km³/year for California (Liu et al., 2022), aligning with our estimate of 12.8 km³/year. Similarly, in the extensively studied Colorado River Basin, we estimate unsustainable water consumption at 3.8 km³/year, consistent with previous estimates of 3–4 km³/year (Richter et al., 2023).

For Pakistan and Algeria, previous work reported water gaps of $37 \pm 12 \text{ km}^3/\text{year}$ and $1.7 \pm 0.6 \text{ km}^3/\text{year}$, respectively (Wada et al., 2010), which align closely with our estimates of $35.8 \text{ km}^3/\text{year}$ and $0.8 \text{ km}^3/\text{year}$. In Egypt, where the water gap has been estimated at around $20 \text{ km}^3/\text{year}$ (Nikiel and Eltahir, 2021), we estimate it to be $18.8 \text{ km}^3/\text{year}$. For Iran, Libya, Saudi Arabia, and Mexico, studies have estimated groundwater depletion-related water gaps of 33.3 km^3 , 2.5 km^3 , 12.5 km^3 , and 11.1 km^3 per year (Dalin et al., 2017), while our estimates are 35 km^3 , 1.7 km^3 , 12.1 km^3 , and 9.6 km^3 per year, respectively.”

References

Jägermeyr, J., Pastor, A., Biemans, H. and Gerten, D., 2017. Reconciling irrigated food production with environmental flows for Sustainable Development Goals implementation. *Nature communications*, 8(1), p.15900.

Liu, P.W., Famiglietti, J.S., Purdy, A.J., Adams, K.H., McEvoy, A.L., Reager, J.T., Bindlish, R., Wiese, D.N., David, C.H. and Rodell, M., 2022. Groundwater depletion in California's Central Valley accelerates during megadrought. *Nature Communications*, 13(1), p.7825.

Scanlon, B.R., Longuevergne, L. and Long, D., 2012. Ground referencing GRACE satellite estimates of groundwater storage changes in the California Central Valley, USA. *Water Resources Research*, 48(4).

Rosa, L., Rulli, M.C., Davis, K.F., Chiarelli, D.D., Passera, C. and D'Odorico, P., 2018. Closing the yield gap while ensuring water sustainability. *Environmental Research Letters*, 13(10), p.104002.

Rosa, L., Chiarelli, D.D., Rulli, M.C., Dell'Angelo, J. and D'Odorico, P., 2020. Global agricultural economic water scarcity. *Science Advances*, 6(18), p.eaaz6031.

Rosa, L., Chiarelli, D.D., Tu, C., Rulli, M.C. and D'Odorico, P., 2019. Global unsustainable virtual water flows in agricultural trade. *Environmental Research Letters*, 14(11), p.114001.

Wada, Y., van Beek, L.P. and Bierkens, M.F., 2012. Nonsustainable groundwater sustaining irrigation: A global assessment. *Water Resources Research*, 48(6).

Dalin, C., Wada, Y., Kastner, T. and Puma, M.J., 2017. Groundwater depletion embedded in international food trade. *Nature*, 543(7647), pp.700-704.

Mekonnen, M.M. and Hoekstra, A.Y., 2020. Blue water footprint linked to national consumption and international trade is unsustainable. *Nature Food*, 1(12), pp.792-800.

Mekonnen, M.M. and Hoekstra, A.Y., 2016. Four billion people facing severe water scarcity. *Science advances*, 2(2), p.e1500323.

Nikiel, C.A. and Eltahir, E.A., 2021. Past and future trends of Egypt's water consumption and its sources. *Nature Communications*, 12(1), p.4508.

Richter, B.D., Lamsal, G., Marston, L., Dhakal, S., Sangha, L.S., Rushforth, R.R., Wei, D., Ruddell, B.L., Davis, K.F., Hernandez-Cruz, A. and Sandoval-Solis, S., 2024. New water

accounting reveals why the Colorado River no longer reaches the sea. *Communications Earth & Environment*, 5(1), p.134.

Finally, the study does not present any new or remarkable results. Water supply and demand changes in the most important regions/basins have been studied in much more detail by others; e.g. on Ganges-Brahmaputra, by Lutz et al, Biemans et al, Wijngaard et al, or for major groundwater depletion regions, by Wada And Bierkens et al.. Increases in water gap of 5% (+1.5) up to 13% in the +3 degree scenario do not appear to be of a magnitude that can't be dealt with through improved practices.

We have acknowledged previous work and better explained the novelty and advance of our study (see answer above). The specific increases in water gaps have not been previously quantified by basin, by country and on a global scale.

It would be incorrect to claim that "increases in the water gap of 5% (+1.5) up to 13% in the +3°C scenario" are not meaningful results. We hope that with our responses and revised manuscript, the novelty and contribution of our work are more clearly highlighted. We addressed this additional important point in the discussion section, which reads:

“The water gap is projected to increase by 5% (22.7 km³/yr) under a 1.5°C warming scenario and 13% (61.7 km³/yr) under a 3°C scenario, highlighting substantial challenges for water sustainability, with the impacts intensifying as temperatures rise. The effects of warming between 1.5°C and 3°C are uneven, with the higher temperature scenario exacerbating issues like groundwater depletion, ecological stress, and unsustainable water use more severely. Even relatively modest increases in the water gap can intensify these problems, putting additional pressure on ecosystems and leading to long-term degradation and difficulty in maintaining freshwater resources. While these projected increases are concerning, they may still be mitigated through improved water management practices and policy interventions.”

REVIEWER COMMENTS

Reviewer #1

The authors have addressed all my comments. I congratulate them for an excellent work.

We appreciate the reviewer's positive feedback and constructive comments.

Reviewer #2

We appreciate the reviewer's critical and constructive comments. Below, we addressed all points raised as requested.

The authors have addressed most of my concerns in detail. I have two further comments:

1. the authors might want to sharpen the novelty of their study still further. I would think most/many research nowadays incorporates future (climate) projections (contrary to what is claimed in L87). And if past research has focused on 'quantifying groundwater depletion' and 'environmental flow reduction' (L83-84) then it has addressed water gaps as defined by the authors, no? That would leave the (global) multi-model approach.

Thank you for pointing this out, we have revised the paragraph in the introduction accordingly:

“Past research has primarily concentrated on quantifying groundwater depletion^{8,11,35-37} and environmental flow reduction^{3,12,30,38-40} on a global scale. Other studies have explored groundwater depletion and unsustainable water use at regional levels⁴¹⁻⁴⁶. However, comprehensive global assessments using multi-model and multi-scenario analyses to evaluate both historical and future water gaps remain lacking. These are missing elements that are essential for informed decision-making and effective adaptation to water scarcity amid uncertainty. This research gap underscores the need for research that can guide policy and bolster resilience in the context of climate variability and growing water demand.”

2. I do believe the results could still be interpreted better, with a discussion of what a 5% or 13% increase in 'water gaps' means, globally, compared to the present-day situation in which there already exist significant water 'gaps'. I do not want to downplay the potential consequences of increases of 5-13% (though with hardly any change in 8 out of 11 largest basins, figure 5), but I wonder if, rather than (only) focusing on the change and the need for comprehensive climate adaptation (as in "... highlighting substantial challenges for water sustainability" (L237) and "These results highlight the need for comprehensive climate adaptation plans that account for changing water availability."(L244)), the study shows that dealing with existing water gaps is

important. This would probably also help adapting to future change in most basins.

The revised text now reads:

“Our findings reveal that under baseline climate conditions, important water gaps already exist, amounting to 457.9 km³ per year, highlighting substantial challenges for water sustainability. Addressing these existing water gaps is crucial, as doing so could likely enhance adaptability to future changes in most river basins. Water gaps are projected to increase by 5% (22.7 km³/yr) under a 1.5°C warming scenario and 13% (61.7 km³/yr) under a 3°C scenario, with the impacts intensifying as temperatures rise (see Table 1 for model specific- and scenario-specific results). The effects of warming between 1.5°C and 3°C are uneven, with the higher temperature scenario exacerbating issues like groundwater depletion, ecological stress, and unsustainable water use more severely. Even relatively modest increases in the water gap can intensify these problems, putting additional pressure on ecosystems and leading to long-term degradation and difficulty in maintaining freshwater resources.

While these projected increases are concerning, they may still be mitigated through improved water management practices and policy interventions. This includes investing in resilient infrastructure, enhancing water storage capabilities^{50,51}, desalination of seawater², reuse of treated wastewater^{2,52,53}, and physical and virtual water transfers to distribute water from areas of abundance to areas of need^{54,55}. By combining these approaches, we can develop a more resilient and sustainable water management system to combat water scarcity.”

Also, interestingly, most increase and higher uncertainty occurs in the smaller 20 of basins (figure 5). Why is that? Do changes simply average out in the larger basins? Could that mean that within those larger basins, there could still be considerable changes in supply and demand?

Figures 3 and 5 use a logarithmic scale, which can make smaller volumes appear to have larger interval bars visually compared to larger volumes. The logarithmic scale allows us to compare the variability among basins and countries with similar water gaps. For example, in figure 3, Italy and Peru – two countries with similar water gaps under baseline conditions – the logarithmic scale allows us to compare their water gaps under each scenario. This would not be possible using a traditional linear scale for water gaps.

In figure 5, while the Java–Timor basin appears to have a larger interval bar, the Ganges-Brahmaputra basin exhibits greater uncertainty. Specifically, under 3°C warming, the Ganges-Brahmaputra basin ranges from 53.6 km³/year to 80.2 km³/year, compared to Java–Timor's range of 1.7 km³/year to 2.4 km³/year. The revised manuscript has Supplementary Table 1-2 with the quantitative results shown in figure 3 and 5 to further help the readers interpret our results. Additionally, all data is provided in the Source Data files and the Zenodo repository.

Supplementary Table 1. The 40 countries with the largest water gaps worldwide. Water gaps are presented under baseline, 1.5°C, and 3°C warmer climate. See Supplementary Dataset for detailed model-specific results.

Km ³ per year	1.5°C warming				3°C warming		
	Baseline	Max	Min	Average	Max	Min	Average
India	124.3	149.8	118.2	133.5	159.4	122.5	139.0
United States	53.8	59.0	54.4	55.9	69.6	52.4	60.8
Pakistan	35.8	43.2	31.6	37.5	66.3	35.1	46.6
Iran	35.0	37.9	33.9	35.7	37.7	34.9	36.8
China	27.2	37.6	22.4	31.0	36.5	30.8	34.1
Iraq	23.8	24.8	23.1	24.1	25.9	23.6	24.7
Egypt	18.8	19.2	18.2	18.7	19.4	18.5	18.9
Uzbekistan	16.6	17.3	15.6	16.5	17.1	15.6	16.4
Saudi Arabia	12.1	12.4	11.6	11.8	16.5	12.2	13.4
Turkmenistan	11.1	11.4	10.2	10.9	11.9	10.6	11.3
Mexico	9.6	9.9	8.2	8.9	11.6	8.8	10.4
Turkiye	6.9	8.7	6.6	7.9	9.8	8.1	9.0
Chile	6.6	6.9	6.5	6.7	7.5	6.8	7.1
Vietnam	6.4	7.1	4.9	5.9	6.9	5.2	6.0
Afghanistan	6.1	6.5	5.4	6.1	7.5	5.8	6.7
Spain	5.7	7.9	6.5	7.3	10.4	7.5	9.1
Sudan	4.6	4.5	3.5	4.1	4.5	3.6	4.0
Syria	3.9	4.5	3.7	4.0	4.7	3.9	4.3
Kazakhstan	3.5	3.8	3.0	3.5	4.0	2.9	3.4
Bangladesh	3.1	3.5	2.7	3.2	7.0	2.6	4.4
Morocco	3.0	4.0	3.2	3.7	4.5	3.8	4.2
Argentina	2.7	3.2	2.6	3.0	3.6	2.8	3.1
Philippines	2.4	2.6	1.6	2.0	2.7	1.7	2.2
United Arab Emirates	2.1	2.1	2.1	2.1	2.2	2.1	2.1
Indonesia	1.9	3.4	0.7	2.0	3.7	1.2	2.1
South Africa	1.9	2.6	1.8	2.2	2.7	1.7	2.4
Libya	1.7	1.8	1.5	1.7	2.0	1.7	1.8
Tajikistan	1.6	1.7	1.5	1.6	1.6	1.4	1.5
Kyrgyzstan	1.5	1.8	1.2	1.6	2.0	1.1	1.5
Greece	1.5	2.3	1.5	1.8	2.7	1.7	2.1
Thailand	1.4	1.7	1.2	1.4	2.3	1.0	1.6
Portugal	1.3	1.5	1.2	1.4	2.1	1.3	1.6
Azerbaijan	1.2	1.5	1.2	1.4	1.6	1.4	1.5
Nigeria	1.2	1.2	0.4	0.8	0.9	0.4	0.6
Australia	1.0	1.7	1.1	1.5	2.1	1.6	1.9
Peru	1.0	1.4	0.9	1.0	1.2	0.7	1.0
Italy	1.0	2.0	1.0	1.4	2.6	1.5	2.3
Canada	0.9	1.2	0.6	0.9	1.3	0.8	1.0
Myanmar	0.9	1.1	0.8	1.0	1.4	0.9	1.1
Russian Federation	0.9	1.0	0.8	0.9	1.4	0.8	1.1

Supplementary Table 2. The 40 major hydrological basins with the largest water gaps worldwide. Water gaps are presented under baseline, 1.5°C, and 3°C warmer climate. See Supplementary Dataset for detailed model-specific results.

Km ³ per year	1.5°C warming			3°C warming			
	Baseline	Max	Min	Average	Max	Min	Average
Ganges - Bramaputra	56.1	70.8	53.5	60.5	80.2	53.6	66.5
Sabarmati	52.6	53.8	47.1	50.5	59.8	47.5	53.3
Tigris - Euphrates	34.1	36.7	33.3	34.8	37.5	35.0	36.1
Indus	28.7	33.1	25.0	29.6	48.8	28.9	36.3
Nile	22.2	22.4	20.5	21.6	22.0	20.8	21.4
Mississippi - Missouri	17.1	22.7	17.2	19.4	28.3	16.8	22.3
Caspian Sea, East Coast	16.6	17.3	15.3	16.5	18.0	15.9	17.1
Central Iran	14.3	15.3	13.8	14.5	15.3	14.0	14.7
Amu Darya	14.2	14.8	13.2	14.1	15.0	13.4	14.3
Columbia and Northwestern United States	13.2	13.9	11.4	12.6	13.5	12.1	12.9
California	12.8	13.3	12.1	12.9	14.0	11.2	13.0
Arabian Peninsula	11.9	11.9	11.3	11.5	16.3	11.9	13.0
Krishna	10.9	14.3	9.9	12.1	12.4	10.8	11.8
Syr Darya	9.9	10.7	8.9	9.9	10.5	8.6	9.7
Ziya He, Interior	8.5	11.1	5.5	8.6	15.3	6.5	10.1
Caspian Sea, South West Coast	7.7	8.8	7.7	8.3	9.0	8.6	8.8
Godavari	5.3	10.1	6.5	7.8	8.9	5.3	7.9
Mediterranean Sea, East Coast	5.3	6.3	5.2	5.8	6.9	6.0	6.5
China Coast	5.2	10.2	3.9	7.1	10.0	5.5	7.8
Mexico, Northwest Coast	4.8	4.7	3.9	4.3	5.3	4.1	4.8
Huang He	4.7	6.8	3.7	4.9	5.8	4.2	5.0
Mekong	4.4	5.0	3.0	4.1	5.4	3.7	4.3
North America, Colorado	3.8	3.8	3.1	3.5	4.1	3.4	3.6
Persian Gulf Coast	3.4	3.5	3.1	3.3	3.8	3.3	3.5
South Chile, Pacific Coast	3.3	3.5	3.2	3.3	3.9	3.4	3.6
North Chile, Pacific Coast	3.2	3.4	3.2	3.3	3.5	3.3	3.4
Africa, North West Coast	3.1	4.0	3.3	3.7	4.5	3.8	4.2
Mediterranean South Coast	2.7	3.4	2.7	3.1	3.9	3.1	3.6
Gulf Coast	2.6	4.4	2.1	3.3	5.5	3.0	4.4
Helmand	2.6	2.7	2.4	2.6	3.1	2.3	2.8
Philippines	2.4	2.6	1.6	2.0	2.7	1.7	2.2
Great Basin	2.1	2.1	1.6	1.9	2.1	1.8	2.0
Red Sea, East Coast	2.1	2.9	2.0	2.4	2.8	2.2	2.5
Gobi Interior	2.1	2.6	1.8	2.2	2.6	1.7	2.2

Eastern Jordan - Syria	2.1	2.2	2.0	2.1	2.2	2.1	2.2
Tarim Interior	2.1	3.2	2.1	2.4	3.6	2.0	2.5
Bo Hai - Korean Bay, North Coast	2.0	3.0	1.6	2.2	2.9	1.7	2.4
Java - Timor	1.9	3.3	0.6	1.9	3.6	1.1	2.0
Rio Grande - Bravo	1.8	1.8	1.3	1.7	2.2	1.5	1.9

REVIEWERS' COMMENTS

Reviewer #2 (Remarks to the Author):

The authors have addressed all my comments.

We thank the reviewer for their work and time in the review of this manuscript.